

# Single-cell data revealed CD14-type and FCGR3A-type macrophages and relevant prognostic factors for predicting immunotherapy and prognosis in stomach adenocarcinoma

Mengling Li[1], Ming Lu[2], Jun Li[1], Qingqing Gui[3], Yibin Xia[3], Chao Lu[3] and Hongchun Shu[4]

[1] Department of General Practice, Shangrao People's Hospital, Shangrao, China
[2] Health Service Center, Shangrao Municipal Health Commission, Shangrao, China
[3] HaploX Genomics Center, Shangrao, China
[4] Digestive System Department, Shangrao People's Hospital, Shangrao, China

Corresponding author
Hongchun Shu, 314290005@qq.com

## ABSTRACT

**Background:** Stomach adenocarcinoma (STAD) exhibits profound tumor heterogeneity and represents a great therapeutic challenge. Single-cell sequencing technology is a powerful tool to identify characteristic cell types.

**Methods:** Single-cell sequencing data (scRNA-seq) GSE167297 and bulk RNA-seq data from TCGA, GTEx, GSE26901 and GSE15459 database were included in this study. By downscaling and annotating the cellular data in scRNA-seq, critical cell types in tumor progression were identified by AUCell score. Relevant gene modules were then identified by weighted gene co-expression network analysis (WGCNA). A prognostic scoring system was constructed by identifying prognostic factors in STAD by Least absolute shrinkage and selection operator (LASSO) COX model. The prognosis and model performance in the RiskScore groups were measured by Kaplan-Meier (K-M) curves and Receiver operating characteristic (ROC) curves. Nomogram was drawn based on RiskScore and prognosis-related clinical factors. In addition, we evaluated patient's feedback on immunotherapy in the RiskScore groups by TIMER, ESTIMATE and TIDE analysis. Finally, the expression levels of prognostic factors were verified in gastric cancer cell lines (MKN7 and MKN28) and human normal gastric mucosal epithelial cells (GES-1), and the effects of prognostic factors on the viability of gastric cancer cells were examined by the CCK8 assay and cell cycle.

**Results:** scRNA-seq analysis revealed that 11 cell types were identified, and macrophages exhibited relatively higher AUCell scores and specifically expressed CD14 and FCGR3A. High macrophage scores worsened the prognosis of STAD patients. We intersected the specifically expressed genes in macrophages subgroups (670) and macrophage module genes (2,360) obtained from WGCNA analysis. Among 86 common genes, seven prognostic factors (RGS2, GNAI2, ANXA5, MARCKS, CD36, NRP1 and PDE4A) were identified and composed a RiskScore model. Patients in low Risk group showed a better survival advantage. Nomogram also provided a favorable prediction for survival at 1, 3 and 5 years in STAD patients. Besides, we found positive feedback to immunotherapy in patients with low
RiskScore. The expression tendency of the seven prognostic factors in MKN7 and MKN28 was consistent with that in the RNA-seq data in addition to comparison of protein expression levels in the public HPA (The Human Protein Atlas) database. Further functional exploration disclosed that MARCKS was an important prognostic factor in regulating cell viability in STAD.

**Conclusion:** This study preliminary uncovered a single cell atlas for STAD patients, and Macrophages relevant gene signature and nomogram displayed favorable immunotherapy and prognostic prediction ability. Collectively, our work provides a new insight into the molecular mechanisms and therapeutic approach for LUAD patients.

## INTRODUCTION

Stomach adenocarcinoma (STAD) threatens the lives and health of at least one million individuals every year and is an urgent public health problem in the world (*Smyth et al., 2020*). Similarly, China had a huge number of STAD patients (*Abbas et al., 2018*; *Yang et al., 2021*). Poor lifestyle habits such as high oil and salt, virus infection or smoking and alcohol abuse in some regions were the leading causative factors of STAD (*Jafari-Sales et al., 2022*; *Yang et al., 2021*). With limited cancer symptoms and unpopular awareness of early screening, many STAD patients progress to advanced stages before being detected (*Hoft, Noto & DiPaolo, 2021*; *Li et al., 2022*). Endoscopy is recognized as the best early screening approach until now and has improved the survival advantage of STAD patients by at least 30% (*Machlowska et al., 2020*). However, lack of awareness of early screening, existing malignant metastasis and treatment resistance (*Chen et al., 2022*; *Choi et al., 2022*), STAD remained the most lethal malignancy with a 5-year survival rate of less than 20% (*Ilic & Ilic, 2022*). Therefore, innovative mining of prognostic and therapeutic biomarkers in STAD was urgent.

Up to data, massive genes models such as oxidative stress and metabolism-related (*Dong et al., 2023*), memory B-cell-corrected miRNA (*Liu et al., 2023*), complement system-related (*Tong et al., 2022*) as well as ferroptosis-relevant (*Zhu et al., 2022*) signature were established hoping to predict the potential risks that patients may face, and help doctors develop more personalized treatment plans, for improving patients' treatment effectiveness and survival rate. However, the heterogeneity of STAD was a huge challenge for treatment, and an in-depth understanding of the mechanisms involved was the key to overcome the treatment tolerance of STAD (*Katzenelbogen et al., 2020*). The emerging single-cell sequencing technology (scRNA-seq) sequenced individual cells to precisely obtain the genetic features and clonal diversity in cells, which allowed the discrimination of tumor heterogeneity by cell clustering (*Yu et al., 2021*). For example, the heterogeneous immune landscape between LUAD and squamous carcinoma were discovered by scRNA seq approach (*Wang et al., 2022*), indicating different immunotherapy methods can be

adopted for these two kinds of lung cancers. In LUAD patients with EGFR mutations, the heterogeneous tumors and immune cell populations in early period were also discovered as a promising diagnostic method for LUAD (*He et al., 2021*). *Shen, Chen & Gao (2023)* constructed a mesenchymal stem cell-relevant model for assessing prognosis and drug sensitivity in STAD patients *via* scRNA-seq approach. Similarly, through integrating scRNA-seq with bulk data, CXCR4 on tumor-infiltrating B cells showed inferior survival outcomes and was considered as a therapeutic targets for STAD (*Su et al., 2023*). However, macrophage relevant study based on scRNA-seq method in LUAD is rare, only one M2 macrophage-related prognostic and immunotherapeutic signature in ovarian metastasis of STAD was published (*Gao et al., 2023*). Therefore, precise identification of macrophage cell types and deeply explore its actions in STAD are meaningful.

Hereby, we portrayed the cell types in STAD as well as the landscape through scRNA-seq data in an attempt to reveal the cell types that threaten STAD prognosis. We found that macrophage subgroup score was higher than other immune cells. Subsequently, using bulk sequencing data we attempted to establish an effective macrophage related prognostic assessment system to precisely assess prognostic and therapeutic features for STAD patients.

## MATERIALS AND METHODS

### Acquisition and processing of single-cell data from STAD patients

This study included a single-cell RNA sequencing project of STAD patients (registration number: GSE167297), derived from GENE EXPRESSION OMNIBUS (GEO, https://www. ncbi.nlm.nih.gov/geo/). To exclude low-quality cellular data, sequencing data quality tests were performed with the following criteria: mitochondrial gene count percentage > 20%, nCount RNA < 500 or nCount RNA > 50,000, nFeature RNA < 200 or nFeature RNA > 4,000. Further, the scDblFinder package (*Germain et al., 2021*) was loaded in the R software (version: 3.6.0) to exclude potential two-cell data under default parameter conditions. After processing, sequencing data from 22,258 cells were included in the follow-up study.

### STAD patient bulk sequencing data download and processing

Transcriptome sequencing data (FPKM format) of 349 STAD tumor tissues and 32 normal samples from The Cancer Genome Atlas (TCGA, https://portal.gdc.cancer.gov/) database were included in this study. A total of 109 and 200 STAD samples sequencing data from the GSE26901 and GSE15459 datasets, and 359 normal stomach tissues transcriptome sequencing data from The Genotype-Tissue Expression (GTEx, https://gtexportal.org/ home/). Samples with incomplete follow-up information and missing survival status in the TCGA cohort were excluded from this study, and samples with survival time less than 10 years were retained. The FPKM data format was converted to TPM format in the SangerBox database (http://sangerbox.com/login.html) (*Shen et al., 2022*) and log2(TPM +1) conversion was performed. Ensembl was converted to gene symbol with multiple gene symbols for extraction of their median values. Samples with incomplete follow-up information and missing survival status in the GSE26901 and GSE15459 datasets were

excluded from this study. TCGA and GTEx cohorts were combined due to the number of normal samples in the TCGA cohort was small. Batch effect removal was performed to exclude errors in data from different sources. The limma package (*Ritchie et al., 2015*) was loaded in the R software, and the removeBatchEffect function was called to perform the batch effect removal process. The merged queue was named TCGA_GTEx-STAD. After processing, 349 tumor and 391 normal samples from TCGA_GTEx-STAD (training set) were included in this study, and 109 and 182 tumor samples from GSE26901 and GSE15459 datasets (validation set) were included in this study.

## ScRNA data analysis

The Seurat package (version: 3.2.3) (*Stuart et al., 2019*) has various functions specifically for processing single-cell data, which were used to process single-cell data from STAD. First, the Seurat package was loaded in the R software, and log-transformed normalization was selected with the criterion: using a size factor of 10,000 molecules per cell. Next, the scRNA data were downscaled and the FindVariableFeatures function was called to select the variance stabilizing transformation (VST) method for ANOVA to obtain the highly variable features (HVGs). Based on the first 3,000 HVGs and calling the ScaleData function to scale the data, RunPCA was called to conduct principal component analysis (PCA). Next, the optimal number of principal components for further dimensionality reduction analysis was selected using the PCElbowPlot function. The Harmony package (*Korsunsky et al., 2019*) was loaded in the R software to remove the batch effect of single cells in the STAD samples. FindNeighbors in the Seurat package was called to select the top 50 principal components to calculate the nearest neighbor distance between cells. The FindClusters function was used to filter the optimal number of clusters of cells and to cluster the cells in the samples.

## Annotation of cell clusters

For the cell clusters that were clustered and obtained, cellular annotation analysis was conducted by marker genes in order to investigate the cell types of these cell clusters. Marker genes were derived from biomarkers found in PD-L1 blockade therapy in gastric cancer by *Li et al. (2023)* and from the SingleCellBase database (http://cloud.capitalbiotech. com/SingleCellBase/) (*Meng et al., 2023*). The type of cell cluster was determined based on marker genes expression. During this process, if a situation was encountered where two or more cells could not be distinguished then the resolution parameter was increased to re-cluster. The Uniform Manifold Approximation and Projection (UMAP) dimensionality reduction analysis of the data was executed by calling the RunUMAP function in the Seurat package. To identify differentially expressed genes (DEGs) in different cell types, the FindAllMarker function of the Seurat package was called to identify genes differentially expressed in them. We chose |log fold change (FC)| > 0.25, $p < 0.05$, minpct = 0.1 as parameters for the analysis. Moreover, based on the identified marker genes in the cell types, the AUCell score for different cell types was calculated by calling the AUCell_calcAUC function in the R package loaded AUCell R package (*Aibar et al., 2017*). AUCell is a new method that allows identifying cells with active gene regulatory networks
in scRNA seq data. The input to AUCell is a gene set, and the output the gene set "activity" (*Stuart et al., 2019*) in each cell. The higher AUCell score indicates that the cells have more active gene regulatory networks.

## WGCNA

WGCNA package (*Langfelder & Horvath, 2008*) was executed in this study for analysis. Initially, the median absolute deviation (MAD) of all genes in TCGA_GTEx-STAD was calculated, and the genes in the top 50% of MAD values were retained. Then the correlations between the remaining genes were calculated and converted into a concatenation matrix. The pickSoftThreshold function of the WGCNA package picked the best soft threshold β based on a power function. Hierarchical clustering analysis was conducted based on the topological matrix in the network, and the dynamic tree shearing algorithm merged the high similarity gene modules. The parameters were set as: minModuleSize = 80, similarity distance < 0.3. For the obtained gene modules, a principal component analysis was conducted with the first principal component as the eigenvector of the module for Pearson correlation analysis with Macrophage traits. For Macrophage-related gene modules, Gene Ontology (GO) and Kyoto Encyclopedia of Genes and Genomes (KEGG) analyses were conducted by loading the clusterProfiler package (*Yu et al., 2012*) in R.

## STAD prognostic system

To effectively assess the prognosis of STAD patients, this study attempted to establish a prognostic assessment system based on scRNA-seq data and RNA-seq data. The overlapping genes in DEGs from Macrophage with other cell types and macrophage-related modules were extracted, loaded into glmnet package (*Simon et al., 2011*) in R software, and trained by LASSO COX model according to the survival status and survival time of STAD patients. The model was trained based on 5-fold cross-validation to select the penalty parameter λ. The model at the best λ was selected as the prognostic scoring system for STAD. The RiskScore was constructed based on the expression data (Expi) of the prognostic factors in the model and the LASSO COX coefficients (βi). The prognostic status of STAD was assessed by the formula RiskScore = Σβi × Expi. The RiskScore values were assessed by plotting KM curves and (ROC curves in TCGA_GTEx-STAD, GSE26901 and GSE15459 according to the median RiskScore groups. In addition, through the Application Programming Interface (API) of the human protein atlas (HPA, https://www. proteinatlas.org/), the HPAanalyze package (*Tran et al., 2019*) to obtain the expression levels of translated proteins of prognostic factors in RiskScore and invoked the hpaVisPatho and hpaVisTissue functions of the HPAanalyze package to visualize the data. We also analyzed the copy number variation of the prognostic factors.

## Correlation between RiskScore and clinical factors

We summarized the age, gender, T stage, N stage, M stage, stage, and grade information of STAD patients in the training set, and performed univariate and multivariate COX analysis with RiskScore. The independent prognostic factors among them were screened to

construct nomogram. The calibration curves at 1, 3, and 5 years were used to determine the ability of nomogram to predict the survival rate of STAD. In addition, to validate the value of nomogram and RiskScore in feeding back STAD prognosis, we plotted the Decision curve to judge their predicted benefit.

## Gene set enrichment analysis of DEGs in RiskScore groups

Limma was used to determine the DEGs in the RiskScore groups ($p$.adj < 0.05 & |log2(FC)| > 1.5), and the significantly upregulated DEGs among them were selected for over Representation Analysis. This process was accomplished by the clusterProfiler package. The HALLMARK pathway (h.all.v2023.1.hs.symbols.gmt) was downloaded from the Human Molecular Signatures Database (MSigDB, https://www.gsea-msigdb.org/gsea/msigdb/collections.jsp). Gene set enrichment analysis (GSEA) and single sample gene set enrichment analysis (ssGSEA) were performed in the RiskScore groups *via* the GSVA package and GSEA software.

## Immunotherapy assessment

First, the landscape of immune cells, stromal cells in STAD samples in the training set was assessed by Estimation of STromal and Immune cells in MAlignant Tumour tissues using Expression data (ESTIMATE) program (*Yoshihara et al., 2013*). Then the immune scores of Myeloid dendritic cell, Neutrophil, B cell, T cell CD4+, T cell CD8+, Macrophage, were extracted in TIMER 2.0 database (http://timer.cistrome.org/) (*Li et al., 2020*). Finally, the TIDE presenting genome-wide expression signatures that measure the level of T cell dysfunction and T cell exclusion in tumors to predict immune checkpoint blockade to clinical response was analyzed in Tumor Immune Dysfunction and Exclusion (TIDE, http://tide.dfci.harvard.edu.), with the higher TIDE score, the bigger chance for immune escape, while the lower TIDE score, the more benefit from immunotherapy (*Jiang et al., 2018*). Also the correlations among Exclusion score, Dysfunction score, Myeloid-derived suppressor cells (MDSCs) scores, and RiskScore were assessed by Spearman correlation.

## Cell culture

Gastric cancer cell lines (MKN7 and MKN28) and human normal gastric mucosal epithelial cells (GES-1) were purchased from the Chinese Typical Culture Reserve Center (Shanghai, China). MKN7, MKN28, and GES-1 cells were placed in DMEM (Gibco, Carlsbad, CA, USA) with addition of fetal bovine serum (Gibco, Carlsbad, CA, USA) and penicillin/streptomycin and were cultured at 37 °C at 37 °F. The culture conditions were 5% $CO_2$. The cell lines were suspended in serum-free cell freeze and stored in liquid nitrogen tanks.

## Quantitative reverse transcription-polymerase chain reaction

TRIzol reagent (Thermo Fisher, Waltham, MA, USA) was employed for total RNA extraction from MKN7, MKN28 and GES1 cell lines, with A260/280 in the range of 1.8–2.0. Using a LightCycler 480 PCR System and FastStart Universal SYBR ®Green Master (Roche, Indianapolis, IN, USA), quantitative reverse transcription-polymerase chain reaction (qRT-PCR) was performed on the extracted RNA from each sample (2 g).

**Table 1 Primer information for target genes.**

| Gene | Forward primer sequence (5′-3′) | Reverse primer sequence (5′-3′) |
| --- | --- | --- |
| RGS2 | CTCTACTCCTGGGAAGCCCAAA | TTGCTGGCTAGCAGCTCGTCAA |
| RNAI2 | GTGCCTCCGGCAACATTGA | GCACGAATCTTTGCAGGGA |
| ANXA5 | GTGGCTCTGATGAAACCCTCTC | GGCTCTCAGTTCTTCAGGTGTC |
| MARCK5 | AGCCCGGTAGAGAAGGAGG | TTGGGCGAAGAAGTCGAGGA |
| PDE4A | CTGCGACATCTTCCAGAACCTC | GCTGGTCACTTTCTTGGTCTCC |
| CD36 | CAGGTCAACCTATTGGTCAAGCC | GCCTTCTCATCACCAATGGTCC |
| NRP1 | AACAACGGCTCGGACTGGAAGA | GGTAGATCCTGATGAATCGCGTG |
| GAPDH | AATGGGCAGCCGTTAGGAAA | GCCCAATACGACCAAATCAGAG |

The cDNA served as a template with a reaction volume of 20 μl (0.5 μl of forward, 2 μl of cDNA template, 10 μl of PCR mixture, and reverse primers, and an appropriate volume of water). The PCR was reacted starting with an initial DNA denaturation phase at 95 °C for 30 s (s), followed by 45 cycles at 94 °C for 15 s, at 56 °C for 30 s, and finally at 72 °C for 20 s. Each sample was analyzed in triplicates. Data from the threshold cycle (*Jimenez-Hernandez et al., 2018*) were obtained using $2^{-\Delta\Delta CT}$ method and standardized to the level of GAPDH. The expression levels of mRNA were compared to normative tissue-derived controls. The primer pair sequences for the genes that were chosen as targets in Table 1 are listed below.

## Cell viability

According to the manufacturer's instructions, cell viability was determined using the Cell Counting Kit-8 test. In 96-well plates, cells from various treatments were grown at a density of $1 \times 10^3$ cells per well. The CCK-8 solution was used at the specified time intervals. The O.D 450 values of each well were determined using a microplate reader (Thermo Fisher, Waltham, MA, USA) following a 2-h incubation at 37 °C.

## Cell cycle

The centrifuge tube was then filled with MKN7, MKN28, and GES1 and progressively dripped with 75% pre-cooled ethanol. The fixed cells were rinsed, and after 30 min, they were stained with PI (2 mg/mL) in PBS containing RNase A (0.1 mg/mL) at room temperature and in the dark. Flowjo software and FACSCalibur flow cytometry (BD Biosciences, San Jose, CA, USA) were used to examine the distribution of cells with various DNA contents at 530 nm excitation wavelength.

## Statistical analysis

Data analysis for the bioinformatics analysis section was implemented with the R software (version: 3.6.0) and sangerbox, and the R package used was obtained through the Bioconductor API (*Gentleman et al., 2004*). ROC curves were plotted by the timeROC package (*Blanche, Dartigues & Jacqmin-Gadda, 2013*). The cellular assay part of the data statistics was realized based on GraphPad Prism 7 (GraphPad Software, La Jolla, CA,

USA). Statistical comparisons between two and multiple groups were respectively performed using Student's t-test and one-way analysis of variance followed by either a Dunnett's test. Prognostic differences in patients in the RiskScore groups were assessed by Log Rank Test in survival analysis. For all analyses, $p < 0.05$ was statistically significant.

## RESULTS

### Cell types in STAD

After screening and processing, sequencing data of 22,258 cells in 14 samples from GSE167297 were included in this study (Figs. S1A and S1B). We selected the top 3,000 HVGs for PCA and the top 50% of PCs for subsequent clustering analysis (Fig. S1C). The batch effect between samples was eliminated before annotation (Figs. S1D and S1E). According to UMAP method clustering, cells in all samples were clearly divided into 26 cell clusters at resolution = 0.8 (Fig. 1A). The 26 cell clusters were defined as 11 cell types according to marker genes expression characteristics (Fig. 1B and Table 2). The marker genes specifically expressed in the 11 cell types were shown by different visualization forms (Figs. 1C, S2 and S3). The number of cells in the samples after quality check and the proportion of annotated cell types are shown in Figs. S1F and S1G. The proportions of B cells, T cells, and Macrophages cell types in the samples and the distribution of cells in clusters were illustrated in Figs. 1D and 1E, with a smaller proportion of Macrophages. Macrophages showed higher AUCell scores, indicating higher cellular activity in Macrophages (Figs. 1F and 1G). These results indicated a greater impact on STAD despite the lower percentage of Macrophages.

### Macrophage-related gene modules

Based on six immune cell (B cell, T cell CD4+, T cell CD8+, neutrophil, macrophage, myeloid dendritic cell) immune scores obtained in TIMER from STAD patients, they were divided into high and low immune score groups based on median values. We found that survival differences were exhibited only in the macrophage groups (Fig. 2A). Macrophage affected survival in STAD patients, and higher macrophage activity was also shown in single-cell data. Macrophage might be a critical factor in regulating the prognosis of STAD. Further, we identified gene modules highly associated with macrophage by WGCNA. The network was biologically significant when the soft threshold β = 8 and the correlation coefficient = 0.85 (Fig. 2B). According to the dynamic shear number procedure, six gene modules were identified (Figs. 2C and 2D). A Pearson correlation analysis was performed with the six gene modules using age, gender, T stage, N stage, M stage, stage, grade, and macrophage as trait vectors. There was a remarkable positive correlation between the pink module and macrophage (r = 0.69, $p < 1e-5$), thus, we defined pink as a macrophage-related gene module (Fig. 2E). Pathway enrichment analysis for genes within the pink module exhibited that these genes were mainly enriched in pathways in cancer, PI3K-Akt signaling pathway, proteoglycans in cancer, regulation of multicellular organismal process as well as regulation of localization. These genes had molecular functions in signaling receptor binding and extracellular matrix structural constituent, and participated in cellular component such as extracellular region part and extracellular space (Fig. 2F).

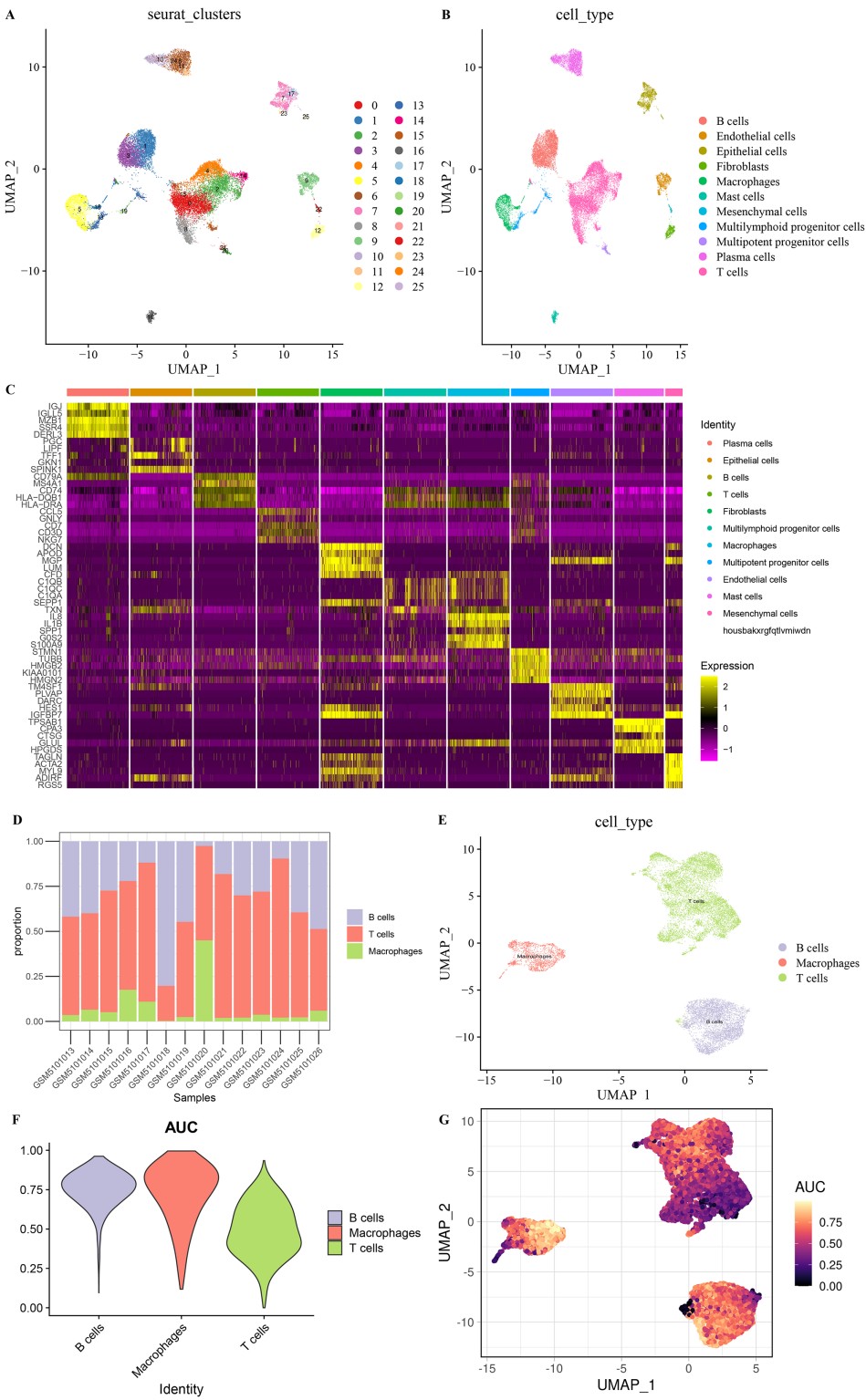

**Figure 1 Eleven cell types in STAD.** (A) 26 cell clusters in STAD. (B) 11 cell types in STAD. (C) Heat map of top five marker genes expression in 11 cell types. (D) Proportions of B cells, T cells, and macrophages in the samples. (E) Distribution of B cells, T cells, and macrophages types in the samples. (F) AUCell score of B cells, T cells, macrophages. (G) AUCell score distribution of individual B cells, T cells, macrophages.

**Table 2 Annotation information of cell types.**

| Cell type | Marker gene | Seurat clusters |
|---|---|---|
| B cells | CD79A, MS4A1 | 1,3,20 |
| T cells | CD3D, CD3E | 0,2,4,8,14,21,24 |
| Macrophages | CD14, FCGR3A | 5 |
| Plasma cells | MZB1, SDC1 | 6,10,11,24 |
| Epithelial cells | EPCAM, KRT19 | 7,17,23,25 |
| Endothelial cells | TM4SF1 | 9,11,12,17,22,25 |
| Fibroblasts | COL1A1, FGF7 | 12 |
| Mast cells | TPSAB1, CPA3 | 16 |
| Multipotent progenitor cells | UBE2S, H2AFX | 12,17,18,20,21,25 |
| Multilymphoid progenitor cells | COTL1, TMSB10 | 0,1,2,3,4,5,7,8,11,12,13,16,17,18,19,20,21,24,25 |
| Mesenchymal cells | ACTA2, AKAP12 | 12,22 |

It indicated that genes within the pink module actively regulated the intercellular communication exchange in STAD.

## RiskScore for predicting STAD prognosis

In single cell analysis, 670 marker genes were significantly differentially expressed in Macrophage cell types *versus* other cells. It was extracted with 86 duplicated genes within the pink module to construct the LASSO COX model (Fig. 3A). The model was observed to be optimal at $\lambda = 0.0389$ under 5-fold cross-validation (Figs. 3B and 3C). RGS2, GNAI2, ANXA5, MARCKS, CD36, NRP1, and PDE4A were the prognostic factors of STAD, among which, GNAI2 and PDE4A were protective indicators, while RGS2, ANXA5, MARCKS, CD36 and NRP1 were risk indicators (Fig. 3D). The prognostic assessment value of RiskScore was judged based on K-M survival analysis and ROC curve in the training set (TCGA_GTEx-STAD) and validation set (GSE26901, GSE15459). We found that the AUC values in 1 year among three dataset were 0.68, 0.67 and 0.72 respectively. The AUC values in 3 years among three dataset were 0.70, 0.71 and 0.71 respectively. The AUC values in 5 years in two validation sets were 0.71 and 0.68 respectively. In short, all AUC values were higher than 0.6 in all three datasets (Figs. 3E, 3G and 3I). We also observed that low RiskScore showed prognostic advantages in all three datasets (Figs. 3F, 3H and 3J). As a result, RiskScore was a trustworthy prognostic system.

## Expression landscape and mutation landscape of prognostic factors

Expression analysis was conducted in order to analyze the expression levels of prognostic factors and protein products in the RiskScore system. First, the protein product expression levels of RGS2, GNAI2, ANXA5, MARCKS, CD36, NRP1, and PDE4A were queried in the HPA database. The expression of the protein encoded by MARCKS gene was lower in STAD tissues than in normal gastric tissues (Fig. 4A). The mRNA levels of RGS2, PDE4A, and CD36 were elevated in STAD tissues, and the mRNA levels of GNAI2, MARCKS, and NRP1 were elevated in normal tissues (Fig. 4B). The expression of seven prognostic factors was also expressed with tissue specificity. RGS2 was specifically expressed in Cytosol,

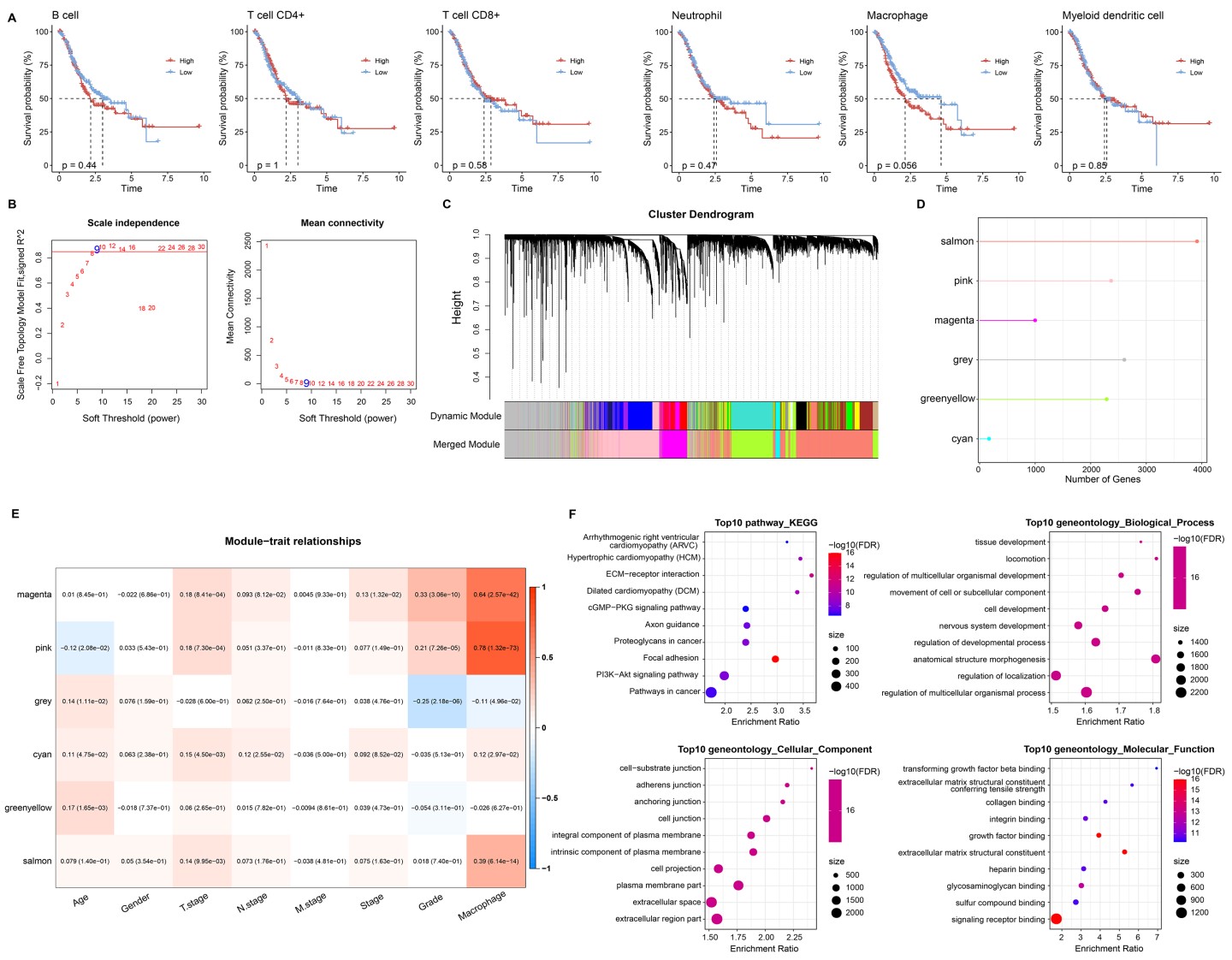

**Figure 2 Macrophage-related gene modules.** (A) K-M curves in B cell, T cell CD4+, T cell CD8+, neutrophil, macrophage, and myeloid dendritic cell immune score groups. (B) Fit indices and network connectivity of scale-free networks with different soft thresholds. (C) Gene module clustering tree. (D) Number of genes in gene-module. (E) Pearson correlation heat map of trait vectors with gene modules. (F) Bubble plots of GO and KEGG results for genes within the pink module.

ANXA5 was specifically expressed in Nuclear membrane, and CD36 was specifically expressed in Golgi apparatus (Fig. 4C). Besides, we found a significant copy number deletion of GNAI2 (Fig. 4D).

## Nomogram based on the composition of RiskScore, stage, and age

We discussed the proportion of patients with different clinical factors in the RiskScore groups, with significant differences between patients in the T stage and grade subgroups (Fig. 5A). As the clinical factors increased, the RiskScore of patients also showed a tendency to increase (Fig. 5B). Univariate and multivariate COX analyses based on clinical factors and RiskScore showed that RiskScore, stage, and age were independent prognostic

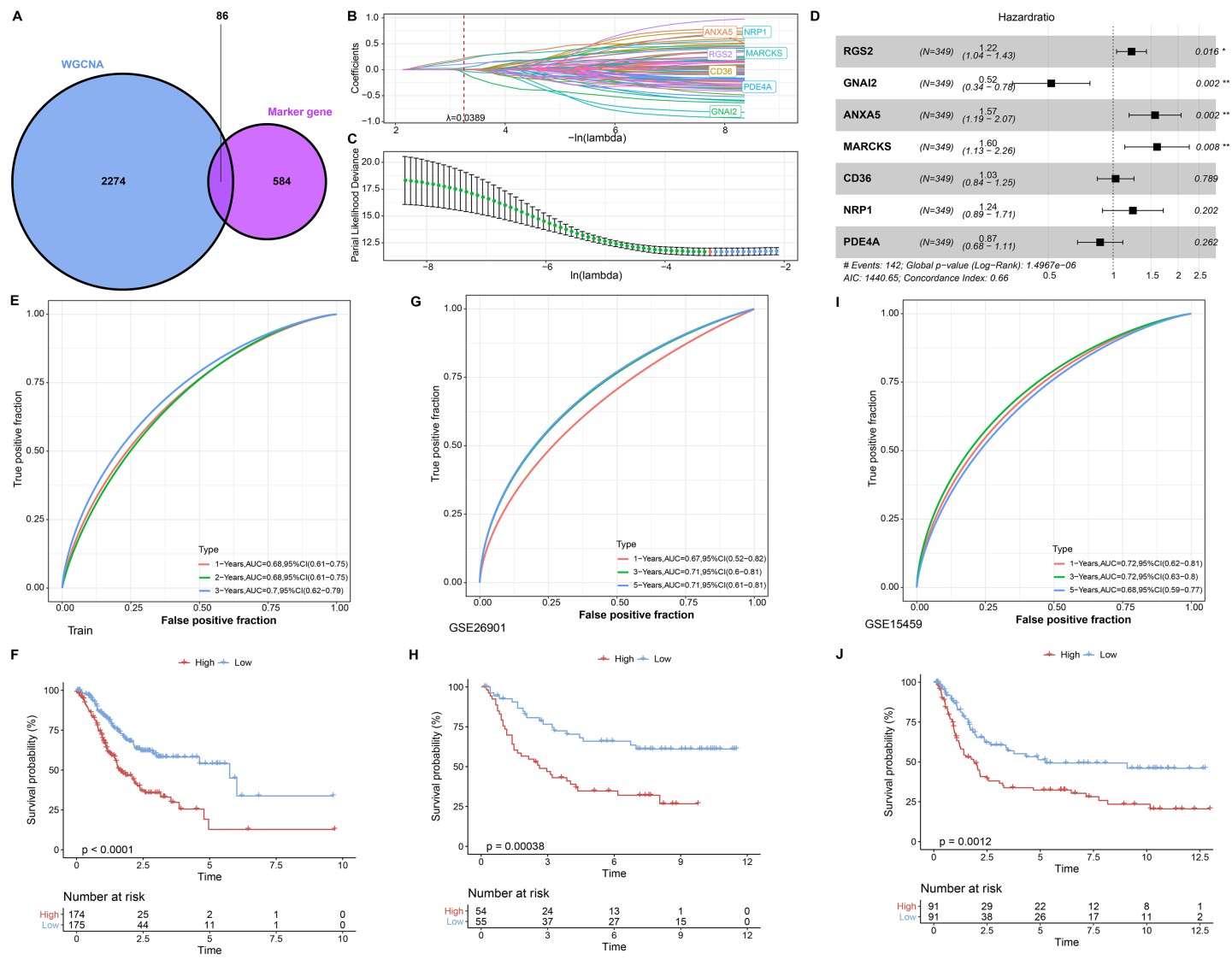

**Figure 3 RiskScore construction and evaluation.** (A) Wayne diagram showing the overlapping genes of pink module genes with differentially expressed marker genes in Macrophage cell types. (B) Trajectory of independent variable coefficients with penalty parameter λ in 5-fold cross-validation in LASSO COX model. (C) Confidence intervals of the penalty parameter λ. (D) Forest plots of the prognostic factors in RiskScore. (E and F) ROC curves of RiskScore and K-M curves in RiskScore groups in the training set. (G and H) In GSE26901, ROC curves of RiskScore and K-M curves in RiskScore groups. (I and J) GSE15459, ROC curves of RiskScore and K-M curves of RiskScore groups. *$p < 0.05$, **$p < 0.01$.

factors for STAD, and all were risk factors (Hazard Ratio > 1, $p < 0.05$) (Figs. 5C and 5D). Therefore, based on RiskScore, stage, and age, we plotted the nomogram used to predict 1-year, 3-year, and 5-year survival of STAD patients (Fig. 5E). The calibration curve showed a good fit between the prediction curve and the actual observed 1-year, 3-year, and 5-year survival curves; therefore, the nomogram was a reliable prognostic prediction tool for STAD (Fig. 5F). Decision curve also further confirmed the strong robustness of nomogram and RiskScore, both of which had higher benefit than the extreme curves (Fig. 5G). Compared to the prognostic value of other clinical factors, the AUC values of

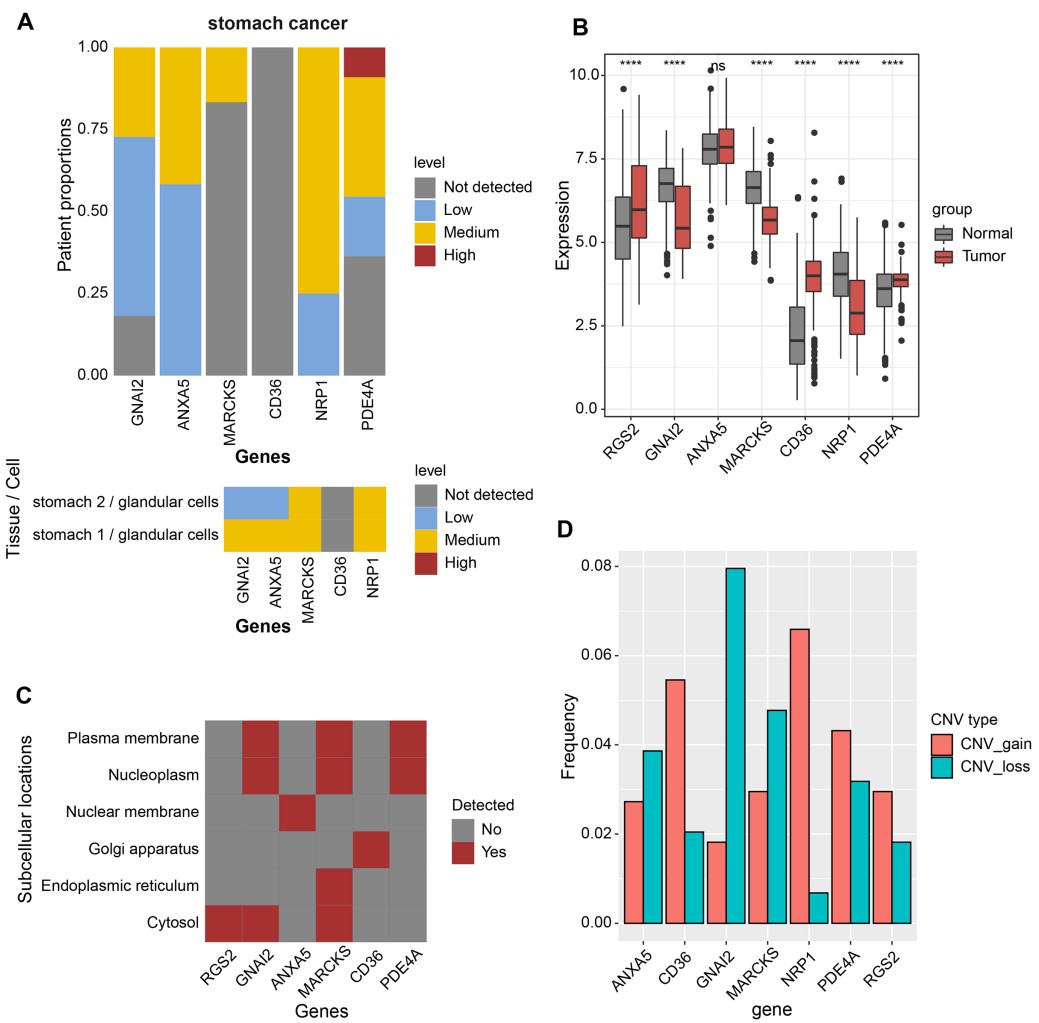

**Figure 4 Expression landscape and mutation landscape of prognostic factors.** (A) Protein expression of prognostic factors in STAD and normal tissues. (B) mRNA expression of prognostic factors in STAD and normal tissues. (C) Prognostic factor expression in cytosol, endoplasmic reticulum, Golgi apparatus, nuclear membrane, nucleoplasm, plasma membrane. (D) Copy number variation statistics of seven prognostic factors. ****$p < 0.0001$; ns, no significance.

nomogram and RiskScore were remarkably superior, showing strong prognostic predictive value (Fig. 5H).

## Biological pathways and immune landscape of differences in RiskScore groups

This study also discussed the biological pathways regulated by DEGs in the RiskScore groups. We found 506 up-regulated DEGs in the high RiskScore group (Fig. 6A), while the number of down-regulated DEGs was minor, so the Over-Representation Analysis was selected. The ORA results showed that the upregulated DEGs were mainly involved in the processes of formation of vascular morphology, extracellular matrix components, and glycosaminoglycan binding (Fig. 6B). The presence of 50 HALLMARK pathways in the RiskScore subgroup was identified in ssGSEA, and we assessed the Spearman correlation

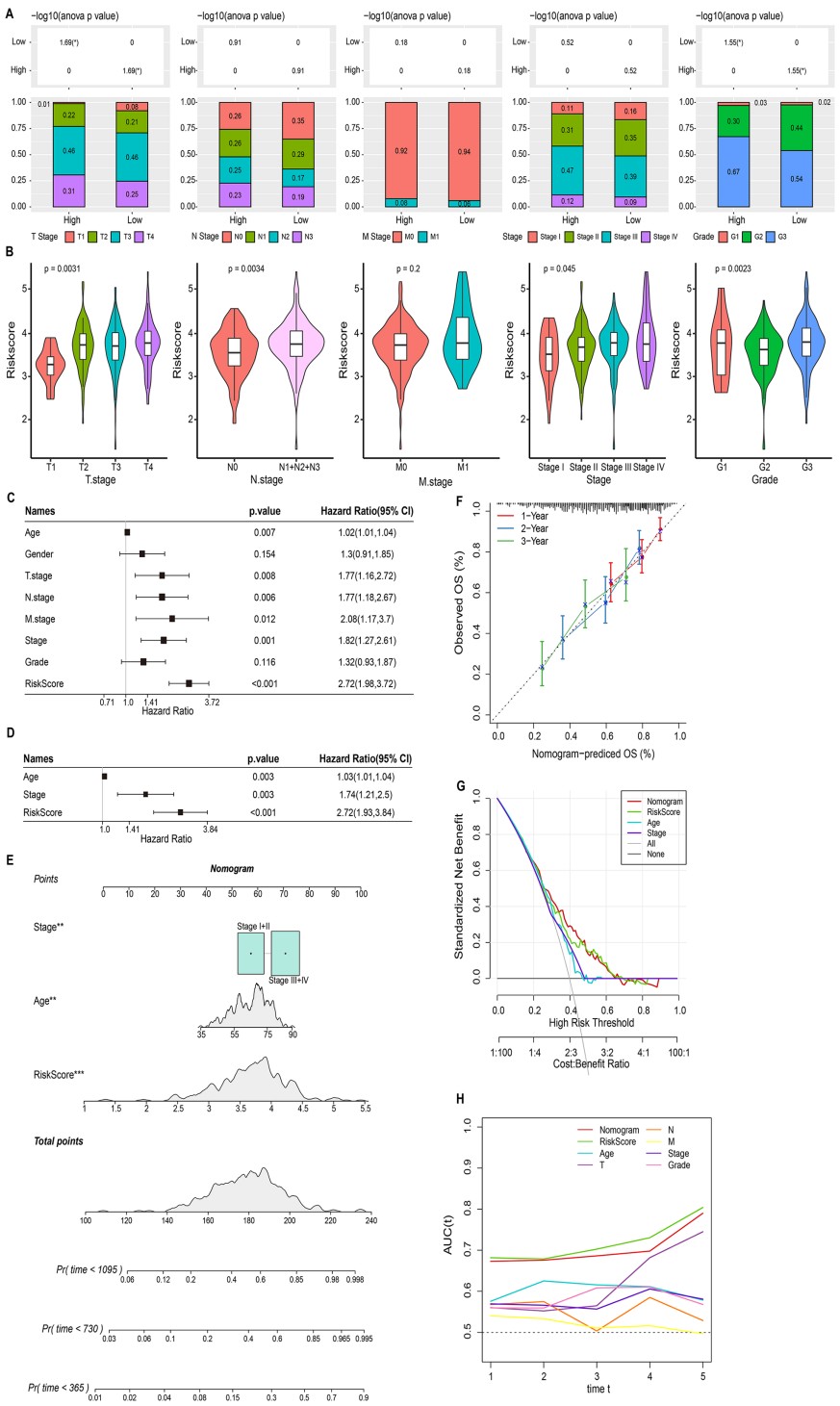

**Figure 5 Nomogram based on the composition of RiskScore, stage, and age.** (A) Proportion of clinical factors subgroup in RiskScore groups. (B) RiskScore among patients in the clinical factor subgroups. (C and D) Univariate and multivariate COX forest plots of clinical factors and RiskScore. (E) Nomogram based on the composition of RiskScore, stage, and age. (F) Calibration curves for 1, 3, and 5 years. (G) Decision curve of RiskScore, nomogram, and clinical factors benefits. (H) ROC curve of RiskScore, nomogram, and clinical factors to predict STAD prognosis.

between them and RiskScore (Fig. 6C). The GSEA results showed that HALLMARK_ TNFA_SIGNALING_VIA_NFKB, HALLMARK_HYPOXIA, HALLMARK_IL6_JAK_ STAT3_SIGNALING, HALLMARK_APOPTOSIS, HALLMARK_APOPTOSIS, and HALLMARK_ANDROGEN_RESPONSE, HALLMARK_MYOGENESIS, HALLMARK_ INTERFERON_GAMMA_RESPONSE, HALLMARK_APICAL_JUNCTION, HALLMARK_COMPLEMENT, HALLMARK_EPITHELEMENT HALLMARK_ EPITHELIAL_MESENCHYMAL_TRANSITION were enabled in high risk group (Fig. 6D). In addition, we assessed the immune landscape of patients in the RiskScore subgroup. ImmuneScore, StromalScore were higher and exhibited higher T cell CD8+, macrophage, myeloid dendritic cell scores in the high RiskScore group (Figs. 6E and 6F). The RiskScore showed a positive correlation with the TIDE score, indicating that the high risk group was more prone to immune escape and less prone to benefit from immunotherapy (Fig. 6G).

## Experimental validation of the reliability of RiskScore models

To validate the RiskScore models, we examined the expression of RGS2, RNAI2, ANXA5, MARCKS, PDE4A, CD36, and NRP1 in MKN7, MKN28, and GES1 cell lines using qRT-PCR (Figs. 7A–7G). The results of qRT-PCR matched the expression levels in RNA-seq data analysis. WB was conducted to detect the expression of MARCKS, which contributed most to the RiskScore models. The results showed that MARCKS expression was elevated in gastric cancer cell lines MKN7 and MKN28 (Fig. 7H). After inhibition of MARCKS in MKN7 and MKN28, decreased cellular activity in MKN7 and MKN28 cells were observed by cell cycle (Figs. 7I and 7J) and CCK8 assays (Figs. 7K and 7L).

## DISCUSSION

STAD continues to be one of the most common causes of cancer death worldwide (Sung et al., 2021). Proven targeted therapeutic markers still lack therapeutic specificity; therefore, there is an urgent need to find novel and effective biomarkers (Choi et al., 2022).

In this study, macrophages were found to be the most active cell type in STAD and specifically express CD14 and FCGR3A. Tumor-associated macrophages (TAMs) were important anti-cancer factors in the anti-tumor immune response. There were numerous biological functions in anti-cancer or pro-cancer responses. On the one hand, TAMs were major regulators of angiogenesis in diffuse STAD, where a high density of M2-type macrophages was the main trigger of the immunosuppressive phenomenon in STAD. On the other hand, TAMs also positively influenced the treatment resistance in STAD patients (Balkwill & Mantovani, 2012; Chung & Lim, 2014; Ge et al., 2018; Joyce & Pollard, 2009). Macrophages were indeed factors affecting the prognosis or treatment of STAD (Wang et al., 2022). Macrophage marker gene CD14, could promote M2-type macrophage abundance (Shi et al., 2021), and FCGR3A+ macrophages might attenuate cytokinesis or the ability to engulf apoptotic cells (Lescoat, Lecureur & Varga, 2021). M2-type macrophages stimulated tumor growth by promoting tumor immunosuppression (Pan et al., 2020). Here, we revealed a specific type of macrophages in STAD, revealing the

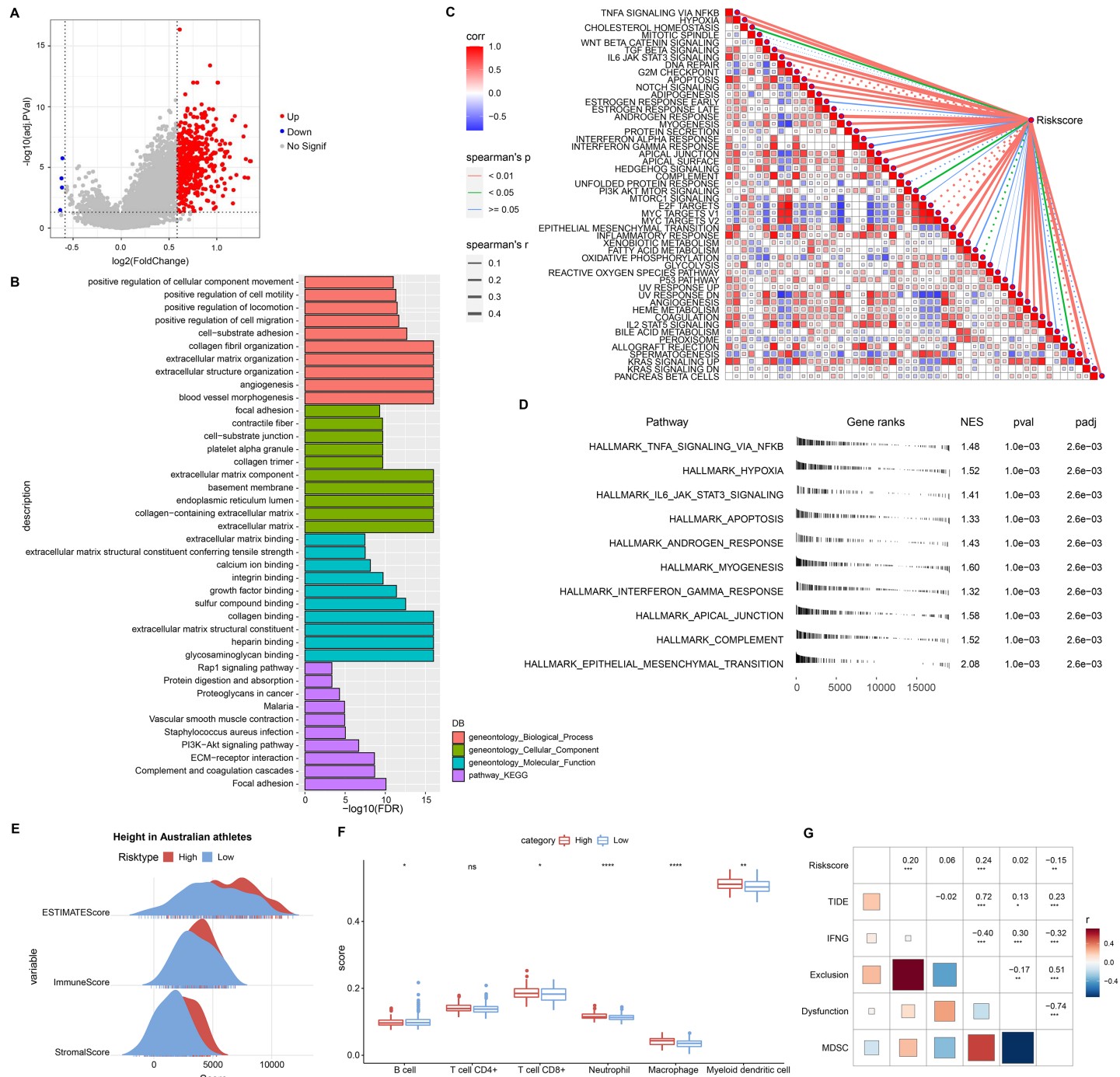

**Figure 6 Biological pathways and immune landscape of differences in RiskScore subgroups.** (A) Volcano map of DEGs in the RiskScore subgroup. (B) Results of ORA analysis of upregulated DEGs. (C) Correlation of RiskScore with HALLMARK pathway. (D) HALLMARK pathway with the most significant difference in the RiskScore subgroup. (E) EATIAMTE results in the RiskScore subgroup. (F) TIMER immune cell scores in the RiskScore subgroup. (G) Correlation of RiskScore with TIDE score. $^*p < 0.05$, $^{**}p < 0.01$, $^{***}p < 0.001$, $^{****}p < 0.0001$, ns, no significance.

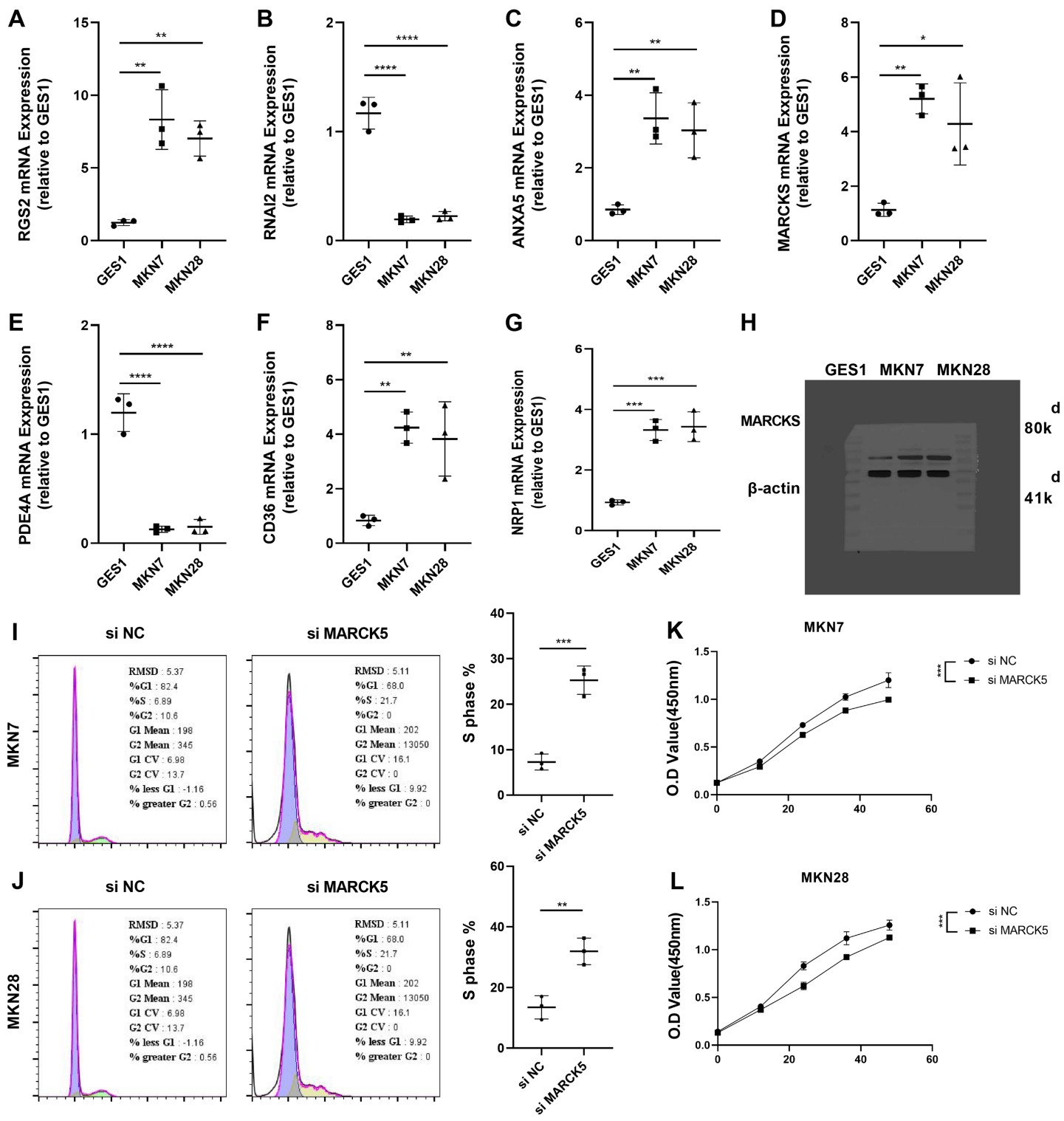

**Figure 7 Experimental validation of the reliability of RiskScore models.** (A–G) The expression of RGS2, RNAI2, ANXA5, MARCK5, PDE4A, CD36, and NRP1 in MKN7, MKN28, and GES1 cell lines was detected and quantified by qRT-PCR. (H) Representative WB results of MARCK5 expression in GES1, MKN7 and MKN28 cell lines. (I and J) Representative cell cycle results following inhibition of MARCK5 expression. (K and L) Representative CCK8 results following inhibition of MARCK5 expression. * $\leq$ 0.05, ** $\leq$ 0.01, *** $\leq$ 0.001, **** $\leq$ 0.0001. The results are presented as mean $\pm$ S.E.M.

heterogeneity of macrophages. And high level of macrophage score affected the poor prognosis of STAD patients.

Macrophage-related genes were mainly enriched in ECM-receptor interaction. Another remarkable feature of cancer patients was the high level of ECM accumulation, and ECM accumulation also worsened the prognosis (*Izzi, Davis & Naba, 2020*). Based on macrophage-related genes, we established a RiskScore prognostic system for LUAD patients. RGS2, GNAI2, ANXA5, MARCKS, CD36, NRP1, and PDE4A were prognostic factors for STAD. Up-regulation of RGS2 expression resulted in a biomarker of delayed value-added and poor prognosis in non-small cell lung cancer, relieving the dormancy of slow-cycling/dormant cancer cells after surgery/chemotherapy and causing tumor recurrence (*Cho et al., 2021*). Coherently, high level of RGS2 was also found in gastric cancer cells, and was closely connected with TIDE, and CD8+ T-cell infiltration in other cancer types (*Yang et al., 2022*), indicating that RGS2 could studied further as a new target for immunotherapy on gastric cancer. *Yu et al. (2022)* showed that GNAI2 facilitated STAD cell growth and migration by promoting PI3K/AKT pathway bioactivity. ANXA5 served as a bridge between the innate and adaptive immune systems, and the ANXA5-phosphatidylserine axis targeted the tumor environment (TME) and delivered chemotherapeutic agents to the TME to kill tumor cells therein (*Woodward, Faria & Harrison, 2022*). Data from scRNA in the ascites of STAD patients indicated that MARCKS marked high plasticity and severe survival in STAD (*Huang et al., 2023*) MARCKS could trigger NF-κB pathway in smoke-relevant lung cancer (*Liu et al., 2021*), and MARCKS-targeting drug (BIO-11006) has displayed preliminary success in clinical trials of lung diseases (*Yadav et al., 2023*). Here, our data further showed that inhibition of MARCKS activity markedly reduced the viability of gastric cancer cell lines. Excessive accumulation of fatty acid molecules in metastatic STAD induced CD63 expression amplifying O-GlcNAcylation signaling levels and stimulating the value-added potential of STAD cells (*Jiang et al., 2019*). NRP1+ lung cancer cells has been ascertained to have tumor-initiating properties (*Jimenez-Hernandez et al., 2018*), and the blockade of NRP1 synergises with anti-PD-1 could strengthen CD8+T-cell proliferation, cytotoxicity as well as tumor inhibition (*Leclerc et al., 2019*). Thus, it was recognized as a novel target for CAR-T immunotherapy in STAD (*Bebnowska et al., 2020*), demonstrating that it is a risk indicator for LUAD. PDE4A was a trigger for metastasis in hepatocellular carcinoma, and inhibition of PDE4A function had the potential to suppress tumor metastasis in patients with hepatocellular carcinoma (*Peng et al., 2018*). Prognostic factors in STAD were mainly associated with tumor cell metastasis, immune cell response, and post-treatment recurrence, and they comprised a RiskScore that took into account important factors in tumor progression. Furthermore, we plotted nomogram, ROC curve and Decision curve also showed that nomogram and RiskScore exhibited excellent robustness and prognostic advantages, and predicting STAD prognosis by our system is of potential clinical value.

Finally, we carried out qRT-PCR analysis of the expression levels of RGS2, GNAI2, ANXA5, MARCKS, CD36, NRP1, and PDE4A in gastric cancer cells. The results were shown to be compatible with the expression levels of 7 prognostic factors in RNA-seq. The results of qRT-PCR indicated that the STAD prognostic factors in the model were

reliable. Besides, MARCKS was a prognostic factor that inhibited the viability of gastric cancer cells. Studies demonstrated that MARCKS was a marker of poor prognosis in highly plastic gastric cancer (*Huang et al., 2023*). However, the functional studies of MARCKS remained poorly characterized, and subsequent focus on its function is still needed.

Our study aimed to identify the specific cell types that affect the prognosis of STAD patients and attempted to find novel cellular targets. An attempt was also made to establish a novel prognostic system, validating their prognostic potential in multiple data. The limitations of present research were as below: although we finally disclosed a single cell atlas and established a RiskScore signature based on microphage-relevant genes, more database and clinical samples are needed to test the accuracy of present findings. Moreover, the mechanism of how model feature genes regulate cancer needs to be further verified in *in vivo/in vitro* assays to enhance their clinical values.

## ABBREVIATIONS

| | |
|---|---|
| **STAD** | Stomach adenocarcinoma |
| **scRNA-seq** | Single-cell sequencing |
| **WGCNA** | Weighted gene co-expression network analysis |
| **LASSO** | Least absolute shrinkage and selection operator |
| **K-M** | Kaplan-Meier |
| **ROC** | Receiver operating characteristic |
| **TME** | tumor microenvironment |
| **GEO** | Gene Expression Omnibus |
| **TCGA** | The Cancer Genome Atlas |
| **GTEx** | Genotype-Tissue Expression |
| **VST** | variance stabilizing transformation |
| **HVGs** | highly variable features |
| **PCA** | principal component analysis |
| **UMAP** | Uniform Manifold Approximation and Projection |
| **DEGs** | differentially expressed genes |
| **FC** | fold change |
| **GO** | Gene Ontology |
| **KEGG** | Kyoto Encyclopedia of Genes and Genomes |
| **API** | Application Programming Interface |
| **HPA** | human protein atlas |
| **MSigDB** | Molecular Signatures Database |
| **GSEA** | Gene set enrichment analysis |
| **ssGSEA** | single sample gene set enrichment analysis |
| **INFG** | interferon gamma |
| **ESTIMATE** | Estimation of STromal and Immune cells in MAlignant Tumour tissues using Expression data |
| **TIDE** | Tumor Immune Dysfunction and Exclusion |
| **MDSCs** | Myeloid-derived suppressor cells |

### Funding

The authors received no funding for this work.

### Competing Interests

The authors declare that they have no competing interests. Qingqing Gui, Yibin Xia and Chao Lu are employed by HaploX Genomics Center.

### Author Contributions

- Mengling Li conceived and designed the experiments, analyzed the data, prepared figures and/or tables, and approved the final draft.
- Ming Lu conceived and designed the experiments, prepared figures and/or tables, authored or reviewed drafts of the article, and approved the final draft.
- Jun Li performed the experiments, authored or reviewed drafts of the article, and approved the final draft.
- Qingqing Gui performed the experiments, prepared figures and/or tables, authored or reviewed drafts of the article, and approved the final draft.
- Yibin Xia analyzed the data, prepared figures and/or tables, and approved the final draft.
- Chao Lu conceived and designed the experiments, analyzed the data, authored or reviewed drafts of the article, and approved the final draft.
- Hongchun Shu performed the experiments, analyzed the data, authored or reviewed drafts of the article, and approved the final draft.

### Data Availability

The data is available at NCBI GEO: GSE167297, GSE26901, GSE15459 and the raw measurements are available at GitHub and Zenodo:

- https://github.com/Shu12q/Raw-data.git.
- Shu12q. (2023). Shu12q/Raw-data: First release of my data (v1.0.0). Zenodo. https://doi.org/10.5281/zenodo.8374644.

### Supplemental Information

Supplemental information for this article can be found online at http://dx.doi.org/10.7717/peerj.16776#supplemental-information.

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
