# Peer review of "Single-cell data revealed CD14-type and FCGR3A-type macrophages and relevant prognostic factors for predicting immunotherapy and prognosis in stomach adenocarcinoma"

_PeerJ, doi:10.7717/peerj.16776_

## Round 0.1 · original submission · Major Revisions

Although both reviewers gave a good overall evaluation of your manuscript, they also raised many issues that require attention. It is recommended to revise and respond to the reviewers' comments one by one.

Reviewer 1 ·

Basic reporting

no comment

Experimental design

no comment

Validity of the findings

no comment

Additional comments

The heterogeneity of Stomach adenocarcinoma poses a great challenge for clinical treatment, which is well addressed by the existence of single-cell analysis technology (scRNA-seq). This study utilized scRNA-seq and bluk RNA-seq analyses to reveal that two macrophage subtypes have key roles in gastric carcinogenesis. The subsequent combination of WGCNA analysis and macrophage subtype-associated differentially expressed genes revealed genes associated with Stomach adenocarcinoma progression and macrophages, and a prognostic model was constructed by multiple types of Cox regression analysis. The accuracy of the model for predicting patient prognostic survival was validated by K-M survival analysis and ROC analysis. Afterwards, the validity of the patient-focused model to predict the genomic mutation profile, clinical profile, biochemical pathway profile, and immune profile of patients was systematically analyzed. Finally, the study validated the reliability of the model characterized genes associated with Stomach adenocarcinoma progression by cellular experiments. In conclusion, the study is based on bioinformatics analysis and supplemented by experimental validation, which is overall logical and experimentally tractable, but the following issues still need to be addressed before publication:
1. What is the significance of mining "CD14-type and FCGR3Atype macrophages" as described in the title of this article? Is this an area not covered by other studies? The title of this article does not make it clear what the value of this study is, so I suggest that it be changed.
2. This article does not say what the analysis of the AUCell score does in Materials and Methods, and does not explain what a high or low AUCell score means for the cell, so it is recommended that a description of this score be added.
3. Please add after the study in line 74-75 a relevant report on the different immune cells available to regulate Stomach adenocarcinoma progression, rather than very abruptly and directly pointing to it as a factor in Stomach adenocarcinoma heterogeneity.
4. Is it possible to illustrate in the Introduction section the reports related to the discovery of cancer heterogeneity by scRNA-seq technology? At the same time, can it be systematically clarified what challenges cancer heterogeneity poses to cancer treatment specifically, e.g., do multiple types of Stomach adenocarcinoma have separate and different treatment regimens? Please add to the original article.
5. The description of Figure 1 is not complete, for example, this article only describes macrophages with high AUCell scores and does not relate to their low level of infiltration, does this mean that even if the AUCell scores are high, their low level of infiltration still has no effect on gastric adenocarcinoma progression? Please explain why.
6. What is the basis for this paper to dig out the pink module by Figure 2, is it only based on its correlation with macrophage characteristics? But here in addition to analyzing the correlation between the module and macrophage, it also involves the correlation between the module and indicators such as age, gender, clinical features, etc.,why ignoring these features to screen the pink module?
7. It is recommended that the description of Figure 3D indicate which characterized genes are prognostic risk factors and which are prognostic protective factors, and it is also recommended that the specific AUC values of the individual ROC curves be elucidated to improve the readability of the full text.
8. The experimental part of this study was mainly to illustrate the reliability of prognostic modeling for predicting the progression of gastric adenocarcinoma, whereas conventional cellular experiments should include the revelation of cell proliferation, migration, invasion, and apoptosis abilities, but why did this study only do experiments related to cell proliferation and cell cycle? Please explain the reasons.
9. Please check the literature reports in the Discussion section on the model-characterized genes mined in this paper, for consistency with the tumor-promoting or anti-tumor trends revealed in this paper and add reasonable hypotheses that link the process of gene modulation of patient prognosis to the other studies in this paper, rather than simply piling on the literature reports.
10. The limitations of this paper need to be elucidated more specifically, such as what kind of experiments are proposed to follow this paper? Which mechanism of cancer regulation by model characterized genes is to be verified? Please add more information to make this paper more complete.

Reviewer 2 ·

Basic reporting

Stomach adenocarcinoma (STAD) is one of the most gastrointestinal malignancies, ranking the fifth in cancer incidence and the third in cancer mortality worldwide. In this study, the authors identified macrophage subtypes and key prognostic genes based on public databases. In addition, the significance of the key genes was verified by cellular experiments. Overall, this is a complete study. However, there are some issues in the manuscript that need to be revised to be able to be published in a journal.
1. In the ABSTRACT section, please note that "STAD" is labeled after “Stomach adenocarcinoma”.
2. In the ABSTRACT section, the names of the databases that are the source of the single-cell and bulk RNA data need to be clearly indicated.
3. In lines 38-42, the authors need to add to the specific methods used. For example, "Algorithms such as ETIMATE and TIDE were used to assess immunotherapy response in patients from different risk groups."
4. In the "Results" and "Conclusion" sections of ABSTRACT, authors should rewrite them. In the "Results" section, the key elements of the study should be clearly expressed and the logic should be clear. In addition, the content of "Conclusion" should clearly mention the clinical and prognostic significance of the study for STAD patients.
5. In the introduction, the authors are overly descriptive about STAD and TME related to it, and there is a serious lack of research and descriptions about STAD and its subtypes and biomarkers, etc. in recent years. In particular, this manuscript focuses around the mechanisms of macrophage subtypes and markers in STAD. However, relevant studies on macrophages in STAD are not addressed at all in this section. This is contradictory. The authors need to rewrite the entire section of the introduction.
6. Please note that all "p"-value in the manuscript need to be changed to "p" with italicized formatting.
7. Please provide figures with more clarity.
8. In line 157, the subheadings are to be bolded and modified.
9. In line 193, additional information and references need to be added about what high and low TIDE scores represent. This is because the authors mention in the results that high and low TIDE scores represent the potential for immune escape.
10. In lines 269-271, please pay attention to the issue of writing format, for example, the case of English letters in a uniform manner.
11. What does " (33538338)" mean in line 342?

Experimental design

no comment

Validity of the findings

no comment

Additional comments

1 This study compares the relationship between different risk scores and clinical factors. However, the relationship between the risk scores and clinical characteristics was not mentioned in Discussion. In addition, it is recommended that the authors link previous studies with prognostic key genes and risk models to explore their clinical relevance.
2 In line345-356, CD14 and FCGR3A as two key markers for macrophage classification should be provided with more references from recent years to provide additional evidence.

---

## Round 0.2 · accepted · Accept

The reviewers have positively acknowledged the novelty, significance, and quality of your study. They noted the thoroughness of your research methodology and the clarity of your writing, which made the manuscript comprehensible and engaging. Both the reviewers do not raise any substantial concerns that warrant further revisions.

Reviewer 1 ·

Basic reporting

no comment

Experimental design

no comment

Validity of the findings

no comment

Additional comments

1、The paper effectively outlines the importance of studying STAD due to its heterogeneity and therapeutic challenges​​.
2、It emphasizes the urgent need for innovative biomarkers and therapeutic targets for STAD, acknowledging the limited treatment options and high mortality rate associated with the disease​​.
3、The study utilizes scRNA-seq and bulk RNA-seq data from multiple databases, implementing rigorous data quality checks and processing methods​​.
4、The application of WGCNA for identifying gene modules and the LASSO COX model for constructing a prognostic scoring system demonstrates a robust analytical approach​​.
5、The identification of macrophage-related gene modules and their association with STAD prognosis is a significant finding​​.
6、The RiskScore model developed shows a strong prognostic value, as evidenced by the AUC values and Kaplan-Meier survival analysis​​.
7、The expression analysis of the prognostic factors and their validation in gastric cancer cell lines add credibility to the findings​​.
8、The study concludes with a significant insight into the molecular mechanisms of STAD and presents a novel approach to prognostic prediction and immunotherapy​​.
9、The manuscript underscores the potential of macrophage-related gene signatures in guiding treatment strategies​​.
Overall, the manuscript presents a valuable contribution to the field of cancer research, particularly in understanding and treating stomach adenocarcinoma. The methodological rigor, significant findings, and potential implications for STAD prognosis and treatment make this a noteworthy study.

Reviewer 2 ·

Basic reporting

no comment

Experimental design

no comment

Validity of the findings

no comment

Additional comments

This study on stomach adenocarcinoma (STAD) is notable for its innovative application of single-cell sequencing technology, identifying key macrophage subtypes (CD14-type and FCGR3A-type) crucial for understanding the tumor microenvironment and immunotherapy response. It skillfully integrates single-cell and bulk RNA sequencing data to identify vital cell types and prognostic factors in STAD progression. The use of statistical methods and the LASSO COX model in developing a prognostic scoring system demonstrates precision and practicality. These findings are significant for STAD clinical treatment, offering valuable biomarkers and informing personalized medicine and immunotherapy strategies. Overall, the study showcases significant innovation and potential impact in scientific methodology and practical application.